# Evaluation of Feasibility on Dental Zirconia—Accelerated Aging Test by Chemical Immersion Method

**DOI:** 10.3390/ma16247691

**Published:** 2023-12-18

**Authors:** Ju-Mei Tian, Wen-Fu Ho, Hsueh-Chuan Hsu, Yi Song, Shih-Ching Wu

**Affiliations:** 1Department of Stomatology of Xiamen Medical College, Engineering Research Center of Stomatological Biomaterials, Fujian Province University, Xiamen 361023, China; 2Department of Chemical and Materials Engineering, National University of Kaohsiung, Kaohsiung 81148, Taiwan; fujii@nuk.edu.tw; 3Department of Dental Technology and Materials Science, Central Taiwan University of Science and Technology, Taichung 40601, Taiwan; hchsu@ctust.edu.tw; 4Department of Dental Technology, School of Medical Technology, Taizhou Polytechnic College, Taizhou 225300, China; songyi@tzpc.edu.cn

**Keywords:** dental ceramic, low-temperature degradation, 3Y-TZP, hydrothermal degradation, dental ceramic

## Abstract

The aim of this study was to investigate the low-temperature degradation (LTD) kinetics of tetragonal zirconia with 3 mol% yttria (3Y-TZP) dental ceramic using two degradation methods: hydrothermal degradation and immersed degradation. To study transformation kinetics, we prepared 3Y-TZP powders. We pressed these powders uniaxially into a stainless mold at 100 MPa. We then sintered the compacted bodies at intervals of 50 °C between 1300 °C and 1550 °C and immersed the specimens at various temperatures from 60 °C to 80 °C in 4% acetic acid or from 110 °C to 140 °C for the hydrothermal method. We used a scanning electron microscope (SEM) to confirm crystalline grain size and used X-ray diffraction to analyze the zirconia phase. As the sintering temperature increased, the calculated crystalline grain size also increased. We confirmed this change with the SEM image. The higher sintering temperatures were associated with more phase transformation. According to the Mehl–Avrami–Johnson equation, the activation energies achieved using the hydrothermal method were 101 kJ/mol, 95 kJ/mol, and 86 kJ/mol at sintering temperatures of 1450 °C, 1500 °C, and 1550 °C, respectively. In addition, the activation energies of the specimens immersed in 4% acetic acid were 60 kJ/mol, 55 kJ/mol, 48 kJ/mol, and 35 kJ/mol, with sintered temperatures of 1400 °C, 1450 °C, 1500 °C, and 1550 °C, respectively. The results showed that a lower sintering temperature would restrain the phase transformation of zirconia because of the smaller crystalline grain size. As a result, the rate of LTD decreased.

## 1. Introduction

Zirconia has been introduced into the dental field since the 2000s. Zirconia ceramic has received an upsurge in favor by dentists and patients not only because of its intriguing properties, such as good mechanical characteristics, high chemical stability, superior esthetic appearance (in terms of transparency and color), and high biocompatibility, but also because of its tremendous clinical applications, such as anterior and posterior single crowns, anterior three- or four-unit fixed partial dentures (FPDs), implants, inlays, onlays, and bridges in dentistry [1,2,3,4,5,6]. Zirconia is available in three different crystalline forms: monoclinic, *m* (room temperature to 1170 °C); tetragonal, *t* (1170–2370 °C); and cubic, *c* (more than 2370 °C) [7,8]. Following sintering, the *t*–*m* phase transformation results in an expansion in volume of about 3–5%, which causes surface compressive stress concentration and results in microcracks in the ceramic material’s structure [9,10,11]. Therefore, stabilizers such as yttria (Y_2_O_3_), calcium oxide (CaO), magnesia (MgO), and ceria (CeO_2_) are added to pure zirconia to maintain the high-temperature tetragonal phase at room temperature. For dental applications, tetragonal zirconia is commonly stabilized with 3 mol% yttria (3Y-TZP) [12].

3Y-TZP has been suggested as an excellent restorative option in the dental field because of its superior flexural strength and its unique phase transformation toughening (PTT) after sintering at high temperatures. The metastable tetragonal phase of 3Y-TZP transforms to the monoclinic phase after it is subjected to external stress, which causes crack propagation. Due to the volumetric expansion of this phase change, the induced compressive stress impedes crack propagation, improving the toughness of 3Y-TZP [10]. Nevertheless, a progressive transformation of the tetragonal phase to the monoclinic (*t*–*m*) phase is caused by the low-temperature degradation (LTD) of zirconia when the material is subjected to a humid environment at temperatures between 100 and 300 °C [13,14,15]. As the oral cavity has similar environmental conditions to those that cause LTD, investigating the impact of LTD on 3Y-TZP materials for dental restorations is vital, specifically regarding their microstructure and mechanical properties.

LTD spreads along the surface of the cavity and gradually moves into the bulk material. This spreading is accompanied by reduced mechanical properties in the material, including long-term wear performance, structural reliability, and phase stability. Since the first reported degradation in 1981, numerous studies have focused on the LTD phenomenon [9,16,17,18]. Numerous short-term catastrophic failures have been attributed to LTD, beginning with water penetration [12,19,20]. ISO 6872 [21], which originally specified dental ceramic materials, has been revised slightly because of the increasingly widespread use of zirconia. In this specification, the chemical solubility of ceramics is determined by immersing the specimens in 4 vol.% acetic acid at 80 °C to measure the weight loss. The chemical solubility of dental ceramic materials in saliva can be accelerated by immersion in 4 vol.% acetic acid at 80 °C.

The oral cavity is an aggressive environment. First, the pH of the oral cavity is constantly subject to alteration by food intake, the amount of plaque, and the composition of stomach acid and saliva. In addition, the corrosion of ceramic materials can reduce mechanical properties and increase surface roughness and plaque adhesion, which can cause tooth wear or discoloration of dental restorations [22]. The stability factors affecting dental materials are the type and composition of the solution. Degradation tests were carried out using distilled water and saliva, and the amounts of ions released between them were different [23]. Using acid (hydrochloric acid or acetic acid) for chemical stability testing can provide faster results for material degradation or long-term chemical stability testing than using distilled water or artificial saliva [24]. In addition, an acid test can simulate the pH of the oral cavity, which is the main reason for selecting 4% acetic acid for the chemical stability test of dental ceramics in ISO 6872:2008. Rijk et al. [25] used five test solutions—tea mixed with tannic acid, artificial saliva, Ringer’s fluid, distilled water, and acetic acid—to test the chemical stability of five dental ceramics. Among all the test solutions, acetic acid was the most aggressive one. Notably, soaking ceramic materials at 80 °C in 4% acetic acid for a week is equivalent to keeping ceramics in artificial saliva at 22 °C for 22 years.

In this study, in addition to the chemical immersion method, we also chose another method. Zirconia test specimens were placed in a suitable autoclave, which was exposed to steam at 134 °C. Following this protocol, we applied a pressure of 0.2 MPa for 5 h. After this process, we cooled the autoclave. The test specimens were removed and dried once cooled. We then conducted subsequent measurements of the monoclinic phase content and mechanical strengths. For convenience, this procedure is commonly called the hydrothermal method. However, dental zirconia typically comes into contact with saliva rather than body fluids. Another limitation is that the hydrothermal method test requires the use of specialized equipment, that is, an autoclave.

Thus, this study aimed to compare the amount of transferred monoclinic phase and activation energy using the immersion method and the hydrothermal method. Furthermore, the study also assessed the influence of crystalline grain size on the possible presence of the monoclinic phase through hydrothermal degradation and immersed degradation. The study findings are expected to provide a theoretical basis for the clinical application of 3Y-TZP materials.

## 2. Materials and Methods

All of the reagents were commercially obtained and used without further purification. 3Y-TZP powder (3 mol% Y_2_O_3_) was purchased from Cerac Inc. (Milwaukee, WI, USA). Acetic acid (≥99.7%) was purchased from Sigma-Aldrich (Sigma-Aldrich Chemie GmbH, Steinheim, Germany; Merck KGaA, Darmstadt, Germany). Distilled water (H_2_O) was prepared in our laboratory. The study design is schematically illustrated in Figure 1.

### 2.1. Specimen Preparation

The 3Y-TZP specimens prepared in this study included commercial zirconia ceramic powder with an approximate particle size of 15–25 μm. The disk-shaped specimens had a diameter of 15 mm and were dry-pressed under a pressure of 100 MPa using a uniaxial hydraulic press from the 3Y-TZP powder. We randomly divided the disk-shaped specimens into six groups, each containing five specimens. We used a high-temperature sintering furnace (F-17-12, purchased from Terder, Taichung, Taiwan) for sintering and placed these specimens inside the furnace. We set the sintering temperatures to 1300 °C, 1350 °C, 1400 °C, 1450 °C, 1500 °C, and 1550 °C for 2 h. The heating rate was 10 °C/min. We allowed the furnace to cool naturally to room temperature after sintering.

### 2.2. LTD Procedures

In this study, we investigated hydrothermal degradation and immersed degradation.

#### 2.2.1. Hydrothermal Degradation

According to ISO 13356:2015 [26], the samples of 3Y-TZP were hydrothermally degraded for different temperature intervals in an autoclave at 110 °C, 120 °C, 130 °C, 134 °C, and 140 °C. The crystal phase was analyzed using an X-ray diffractometer (XRD, Shimadzu XRD-6100 diffractometer, Shimadzu corporation, Tokyo, Japan). We investigated the phase change and kinetics of 3Y-TZP at different temperatures.

#### 2.2.2. Immersed Degradation

According to ISO 6872:2008 [27], we immersed the samples of 3Y-TZP in 100 mL of acetic acid solution (4% (*v*/*v*)) for 16 h at constant temperatures of 60 °C, 65 °C, 70 °C, 75 °C, and 80 °C. We washed the excess acid with deionized water.

### 2.3. XRD Analysis

We analyzed the progress of *t*–*m* transformation and phase identification using XRD with Cu-Kα radiation (λ = 1.5418 Å) in the 2*θ* range of 25–37° with a step size of 0.02° and a scan speed of 1°/min. The XRD accelerating voltage and emission current were 30 kV and 15 mA, respectively. We used the following equation, introduced by Garvie and Nicholson, to determine the monoclinic phase content (%) on the surfaces, X_Ma_ [28,29,30]:(1)$$X_{m}=\frac{I_{m}\left(\bar{1}11\right)+I_{m}\left(111\right)}{I_{m}\left(\bar{1}11\right)+I_{m}\left(111\right)+I_{t}\left(111\right)}$$
where $I_{m}\left(111\right)$, $I_{m}\left(\bar{1}11\right)$, and $I_{t}\left(111\right)$ are the relative intensities of the monoclinic (111) (2*θ* = 31.2°), monoclinic $\left(\bar{1}11\right)$ (2*θ* = 28°), and tetragonal phases (2*θ* = 30°), respectively [31].

Following the XRD method, originally presented by Korsmač et al. [32,33], we calculated the thickness of the specimens’ transformed surface layer—that is, the transformed zone depth (TZD). We assumed that all the tetragonal grains were transformed into monoclinic symmetry within the transformed surface layer as follows:(2)$$TZD=\frac{sin\theta }{2\mu }ln\frac{1}{1-Xm},$$
where the absorption coefficient *µ* is 0.0642, the angle of reflection *θ* is 15°, and *Xm* is the relative monoclinic phase content obtained from the XRD analysis.

### 2.4. Scanning Electron Microscopy Analysis

The test pieces were first prepared by polishing, and then the polished surface of the sintered samples was thermally etched at 1200 °C for 15–60 min with a heating rate of 20 °C/min. Finally, the samples were gold-plated. We investigated the microstructure of each specimen under a vacuum environment using a scanning electron microscope (SEM) (JSM-7401F, JEOL, Tokyo, Japan).

### 2.5. Mechanical Properties

A microhardness tester (FM-300e type c, FUTURE-TECH CORP, Kawasaki, Japan) was applied to the surface of the test sheet at a load of 2 kg and maintained for 15 s.

### 2.6. Crystalline Grain Size Calculation

The crystalline grain size was calculated according to the intercept method of ASTM E112-2013 (R2021) [34] as follows: crystalline grain size = (total length measured/number of grains passed) × (1/magnification).

### 2.7. LTD Kinetics

The kinetics of hydrothermal degradation and immersed degradation were calculated using Mehl–Avrami–Johnson (MAJ) laws as follows [20,35,36]:(3)$$f=1-exp[-\left(b\cdot t{)}^{n}\right]$$
where *f* is the content of the *m* phase, *t* is the time, *n* is a constant (the MAJ exponent) relating to nucleation and growth, and *b* is a parameter dependent on temperature, which follows the Arrhenius equation:(4)$$b=b_{0}\;exp\left[-\frac{Q}{RT}\right]$$
where *b*_0_ is a constant, *R* is the ideal gas constant (as 8.31446 [J mol^−1^ K^−1^]), *Q* [kJ/mol] is the apparent activation energy, and *T* is the absolute temperature.

## 3. Results

### 3.1. Sample Characterization

The crystalline grain size, calculated according to the linear intercept method presented by ASTM E112-2013 (R2021) [34], increased with the increase in sintering temperature, as shown in Figure 2, which is confirmed by the SEM image (Figure 3). We calculated the crystalline grain size as 215 ± 3 nm, 232 ± 7 nm, 283 ± 5 nm, 355 ± 9 nm, 451 ± 9 nm, and 555 ± 6 nm for sintering temperatures of 1300 °C, 1350 °C, 1400 °C, 1450 °C, 1500 °C, and 1550 °C, respectively.

Representative monoclinic phase content diagrams of hydrothermal degradation are shown in Figure 4. In every sintering temperature group other than the 1350 °C and 1300 °C groups, we detected the formation of the monoclinic phase. The XRD analysis for the 1400 °C, 1450 °C, 1500 °C, and 1550 °C groups indicated monoclinic phase content depending on the sintering temperature, although that of 1400 °C was much less than the others. For instance, in the case of 140 °C hydrothermal degradation, the monoclinic phase content after 24 h was 82%, 83%, and 84% for the 1450 °C, 1500 °C, and 1550 °C sintering temperature groups, respectively.

Representative monoclinic phase content diagrams of immersed degradation are reported in Figure 5. For instance, in the case of 80 °C immersed degradation, the monoclinic phase content after 672 h was 72%, 76%, and 78% for the 1450 °C, 1500 °C, and 1550 °C sintering temperature groups, respectively. For the 1400 °C sintering temperature group, the monoclinic phase content showed an obvious increase with the lengthening of the soaking time (Figure 5e). In addition, specimens in the 1350 °C group produced a small quantity of monoclinic phase under immersed temperatures of 75 °C and 80 °C after immersion for 500 h (Figure 5d,e). The XRD results indicated no detectable monoclinic phase for the 1300 °C group.

### 3.2. Mechanical Properties

Figure 6 presents the microhardness values after immersion at different sintering temperatures. Before the low-temperature decay test, the hardness value of the 3Y-TZP test specimen was the lowest at approximately 990 ± 21 Hv at 1300℃. With the increase in sintering temperature, the hardness value also gradually increased to 1305 ± 28 Hv at 1550 °C. As shown in Figure 6e, the hardness value of sintering dropped the fastest to 1187 ± 44 Hv at 1550 °C, followed by 1195 ± 30 Hv at 1500 °C, and the phase variables of the two gradually increased with the soaking time. After soaking in an acetic acid solution at 80 °C for 28 days, we observed that the hardness changes at 1300 °C and 1350 °C were not greater than 974 ± 41 Hv and 1176 ± 48 Hv, respectively, while the hardness decreased slightly to 1195 ± 26 Hv and 1209 ± 21 Hv at 1400 °C and 1450 °C, respectively.

The measurement of the depth of the monoclinic crystal transformation is shown in Figure 7. The longer the low-temperature decay time and the greater the content of the monoclinic crystal phase, the deeper the measured phase transition layer. In Figure 7e, we display the scenario of immersion in acetic acid at 80 °C. The phase change layers sintered from 1350 °C to 1550 °C were 0.09 ± 0.01, 1.69 ± 0.2, 2.59 ± 0.11, 2.91 ± 0.17, and 3.13 ± 0.17 µm, respectively. This experimental result is also consistent with that of Elshazly et al. [37]. When the material had more monoclinic crystal layers, the hardness value also decreased.

### 3.3. Degradation Kinetics

The MAJ equation could be fitted onto the experimental data by plotting $\ln\left(\ln\left(\frac{1}{1-f}\right)\right)$ as a function of ln(*t*) for hydrothermal degradation and immersed degradation, as shown in Figure 8 and Figure 9, respectively. The *n* and *b* parameters were determined from the slope and intercept of the linear correlations for hydrothermal degradation and immersed degradation, as reported in Table 1. The value of *n* reflects nucleation and growth behaviors [38,39]. According to dimensionality, it is possible to classify nanocrystalline materials into the following categories: (1) zero-dimensionality is atom clusters or cluster assemblies (*n* < 1); (2) one-dimensionality reflects linear growth (1 ≤ *n* < 2); (3) two-dimensionality modulates planar overlayer growth (2 ≤ *n* < 3); and (4) three-dimensionality reflects equiaxed stereoscopic structure growth (3 ≤ *n* < 4). In the current study, whatever LTD tests, the higher sintering temperature correlates with the lower *n* values [30,40,41].

## 4. Discussion

According to these results for SEM and XRD characterization, the crystalline grain size increased as the sintering temperature increased, and the phase variable also increased. Kosmač and Kocjan [42] specified that the phase transition rate of a high sintering temperature was faster than that of a low sintering temperature. We compared the monoclinic content plots according to the hydrothermal and immersion methods and found that temperature was responsible for the faster phase transition rate of the hydrothermal method. Ganor, Mogollón, and Lasaga [43] indicated that temperature was an important factor that influenced the reaction mechanism of substances in solution. According to the Van’t Hoff rule and Galwey and Brown [44], the reaction rate increased about twice for every 10 °C increase in the temperature. As the low-temperature decay continued, however, the monoclinic phase reached saturation, and then the phase content no longer increased. As shown in Figure 4, the monoclinic phase content of 3Y-TZP reached more than 80%, and in Figure 5e, it was more than 75%, at which point the phase variable gradually decreased. In this study, the temperature of the hydrothermal method was higher than the immersion method, and the rate of transition from the tetragonal to the monoclinic phase occurred relatively quickly. Similarly, when the phase variable reached saturation, the phase transition did not increase significantly with the extension of the low-temperature decay time.

According to the mechanical property analysis, the experimental results indicated that the higher the sintering temperature, the higher the mechanical properties of the material. After soaking, it was also found that a higher sintering temperature of the test piece was correlated with a lower hardness value with the extension of time. This result is consistent with that of Borchers et al. [45]. According to the research results of Lorente et al. [46], a large number of phase transitions for a material results in a decrease in hardness, which is consistent with our results.

According to the *n* values of the hydrothermal degradation, the planar overlayer growth was the predominant step of the *t*–*m* transformation for the sintering temperatures of 1450 °C, 1500 °C, and 1550 °C. In contrast, growth was complicated for immersed degradation. The three-dimensionality equiaxed growth was predominant for the sintering temperatures of 1400 °C and 1450 °C, and the two-dimensionality overlayer growth was predominant for the sintering temperatures of 1500 °C and 1550 °C. According to Gremillard, Chevalier, Epicier et al. [47], the *n* value exhibited changes at different sintering temperatures, and the higher sintering temperatures resulted in larger grains and faster nucleation. Therefore, the growth rate of the nucleus was also affected by the size of the grain.

By fitting ln(*b*) as 1/*T*, we also estimated apparent activation energies for hydrothermal degradation and immersed degradation (see Figure 10 and Figure 11, respectively). As reported in Table 2, we used the slope of the linear correlations for hydrothermal degradation and immersed degradation and determined the activation energies. According to the experimental results, we inferred that the higher the sintering temperature, the lower the activation energy required for 3Y-TZP to transform from the tetragonal phase to the monoclinic phase.

The activation energy value of the immersion method was lower than that of the hydrothermal method, which may have been related to the immersion solution of acetic acid. The pH value is an important factor affecting the phase change. Lasaga [48] believed that the activation energy would be reduced when organic acid solutions are used and kept in a weakly acidic environment (pH = 3–5). Lawson [49] used kaolinite to test the activation energy at different pH values. The results showed that the activation energy was 6.6 kcal/mol in a pH 2 solution, 9.4 kcal/mol when the pH value increased to 4.2, and 15 kcal/mol when the pH value increased to 7. At pH 8.4, the activation energy decreased gradually to about 7 kcal/mol. Therefore, the difference in activation energy between the hydrothermal method and the immersion method was indeed related to pH value.

In addition to acidic solutions, Lasaga and Gibbs [50] noted that temperature was another important factor affecting the activation energy. A faster reaction speed of the material resulted in a higher heat content being absorbed by the material, and a higher activation energy was also correlated with a higher temperature. The experimental temperatures of the hydrothermal method in this study were between 110 °C and 140 °C. The temperatures of the immersion method were between 60 °C and 80 °C. As a result, the phase transition reaction speed of the hydrothermal method was fast, and the activation energy was higher. The immersion method had a lower phase transition rate and a lower activation energy.

## 5. Conclusions

Based on the results of this study and given its limitations, we draw the following conclusions:Crystalline grain size increased as the sintering temperature increased, and the phase variable also increased.By comparing the hydrothermal method with the immersion method, we found that the higher the temperature of the degradation test, the faster the phase transition rate of 3Y-TZP.The *t*–*m* transformation rate of hydrothermal degradation was significantly larger than that of immersed degradation. The activation energies of hydrothermal degradation were also significantly higher than those of immersed degradation.According to the MAJ calculation, the higher sintering temperature was correlated with the lower energy of the *t*–*m* phase transformation.

## Figures and Tables

**Figure 1 materials-16-07691-f001:**
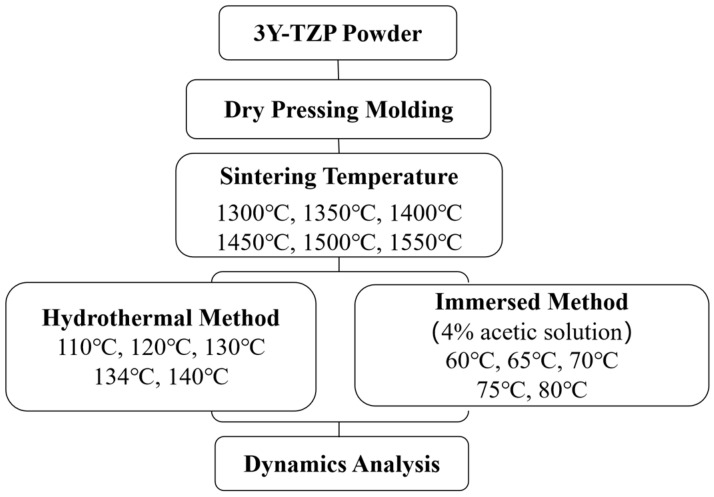
Experimental study setup. 3Y-TZP, tetragonal zirconia with 3 mol% yttria.

**Figure 2 materials-16-07691-f002:**
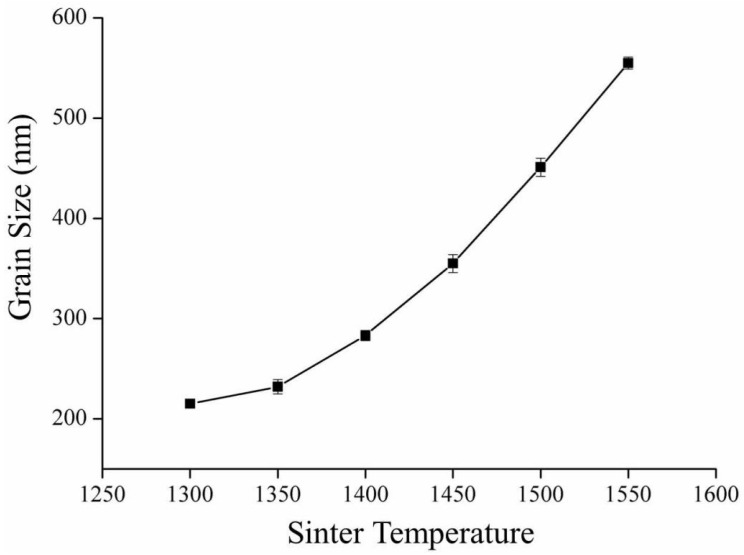
Crystalline grain size variation with sinter temperature for 3Y-TZP.

**Figure 3 materials-16-07691-f003:**
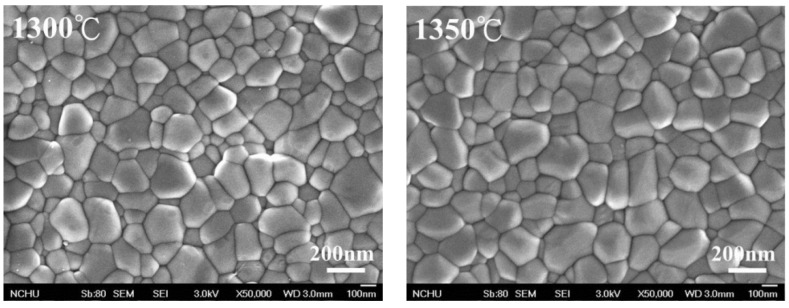
SEM images after sintering through different temperatures for 3Y-TZP. SEM, scanning electron microscope.

**Figure 4 materials-16-07691-f004:**
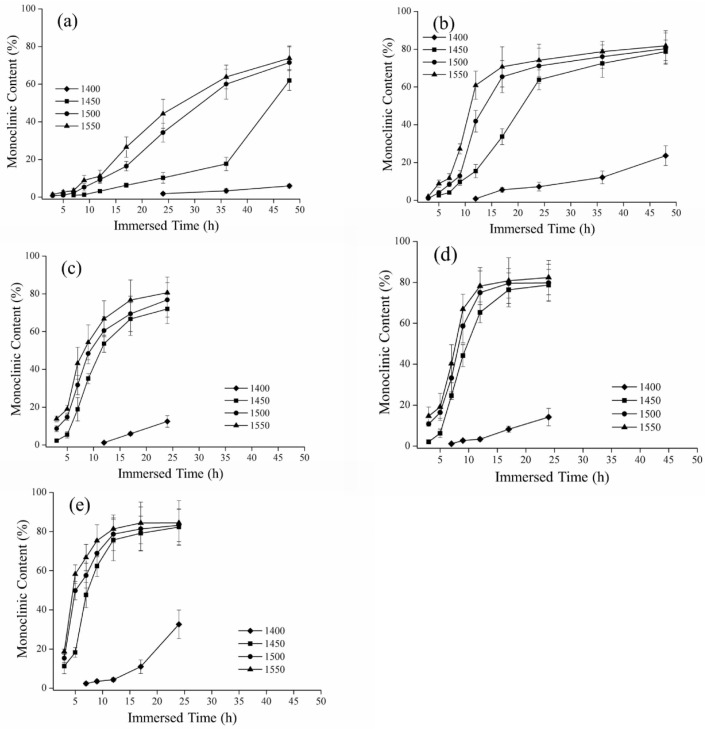
Relationship between hydrothermal degradation time and monoclinic content: (**a**) 110 °C; (**b**) 120 °C; (**c**) 130 °C; (**d**) 134 °C; and (**e**) 140 °C.

**Figure 5 materials-16-07691-f005:**
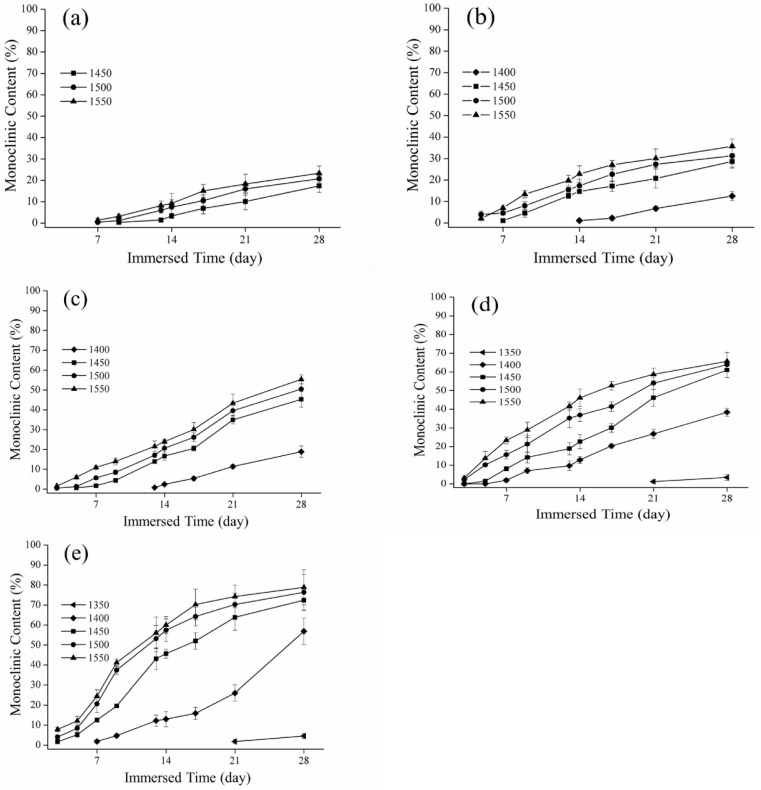
Relationship between immersed time and monoclinic content: (**a**) 60 °C; (**b**) 65 °C; (**c**) 70 °C; (**d**) 75 °C; and (**e**) 80 °C.

**Figure 6 materials-16-07691-f006:**
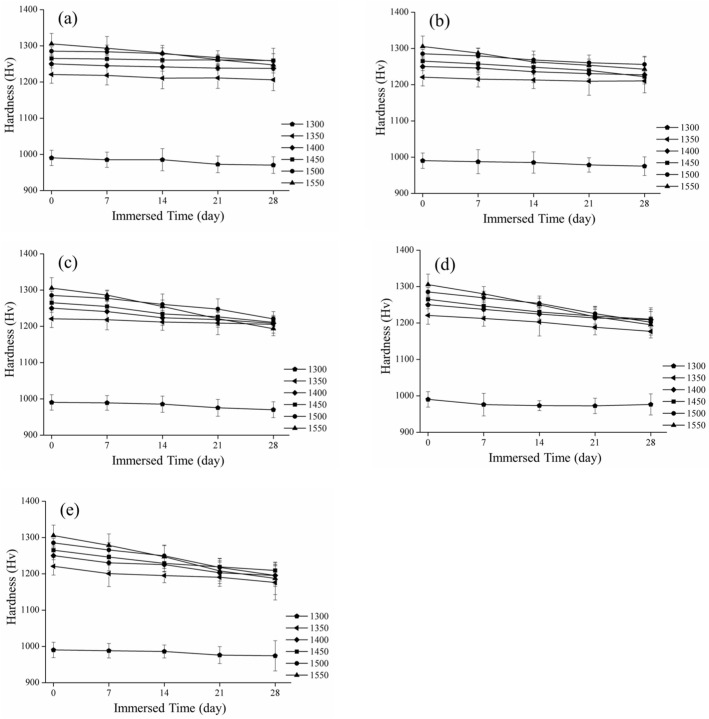
Spot of the hardness and immersed time: (**a**) 60 °C; (**b**) 65 °C; (**c**) 70 °C; (**d**) 75 °C; and (**e**) 80 °C.

**Figure 7 materials-16-07691-f007:**
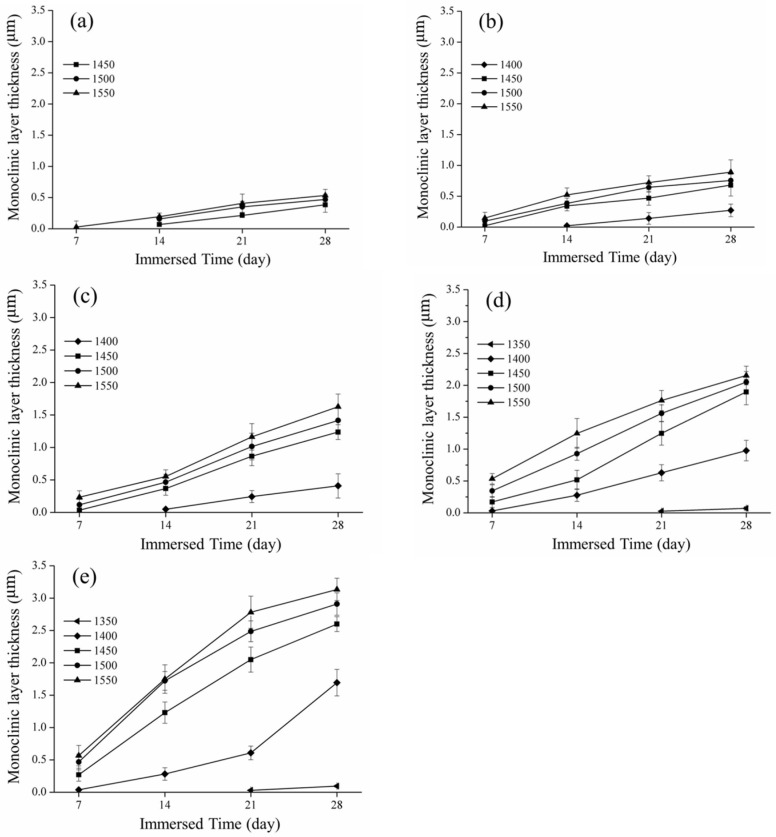
Spot of the phase transition depth and immersed time: (**a**) 60 °C; (**b**) 65 °C; (**c**) 70 °C; (**d**) 75 °C; and (**e**) 80 °C.

**Figure 8 materials-16-07691-f008:**
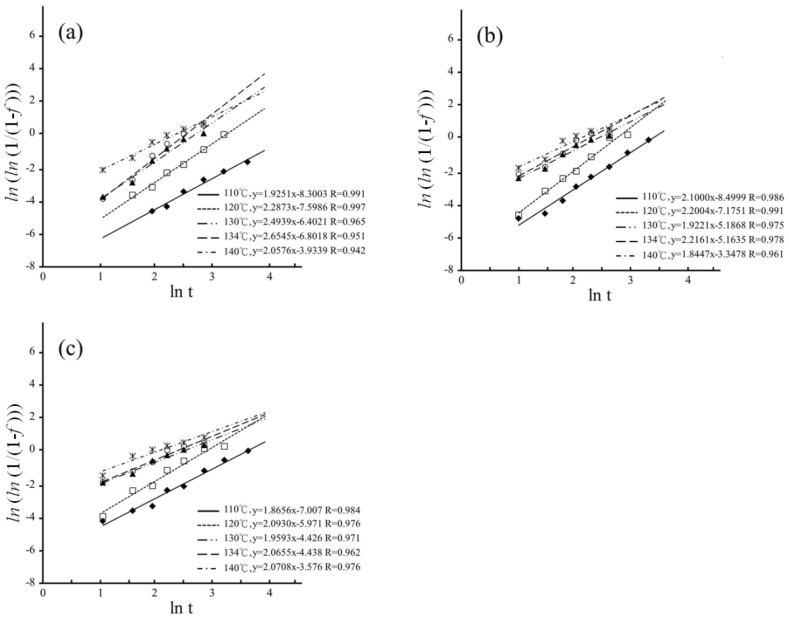
Plotting $\ln\left(\ln\left(\frac{1}{1-f}\right)\right)$ as a function of ln(t) for hydrothermal degradation: (**a**) 1450 °C; (**b**) 1500 °C; and (**c**) 1550 °C. The different symbols represent the different temperature.

**Figure 9 materials-16-07691-f009:**
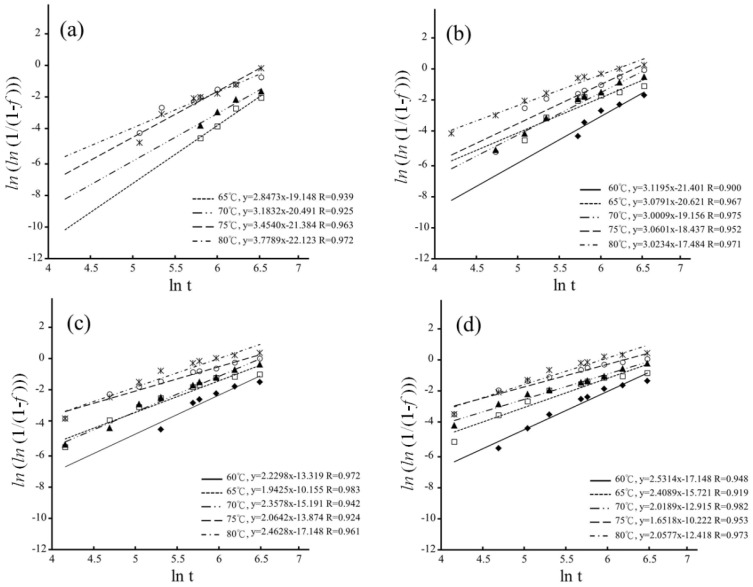
Plotting $\ln\left(\ln\left(\frac{1}{1-f}\right)\right)$ as a function of ln(t) for immersed degradation: (**a**) 1400 °C; (**b**) 1450 °C; (**c**) 1500 °C; and (**d**) 1550 °C. The different symbols represent the different temperature.

**Figure 10 materials-16-07691-f010:**
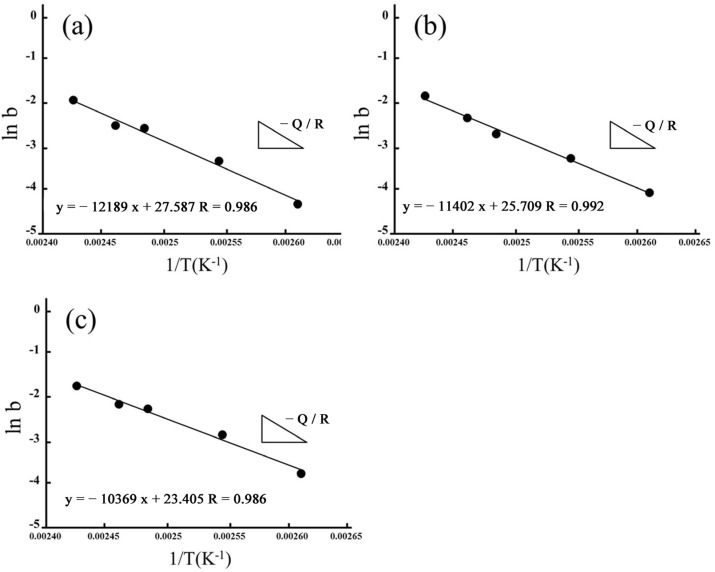
Apparent activation energies of the phase transformation for hydrothermal degradation. Arrhenius plots linearly fitted for (**a**) 1450 °C; (**b**) 1500 °C; and (**c**) 1550 °C.

**Figure 11 materials-16-07691-f011:**
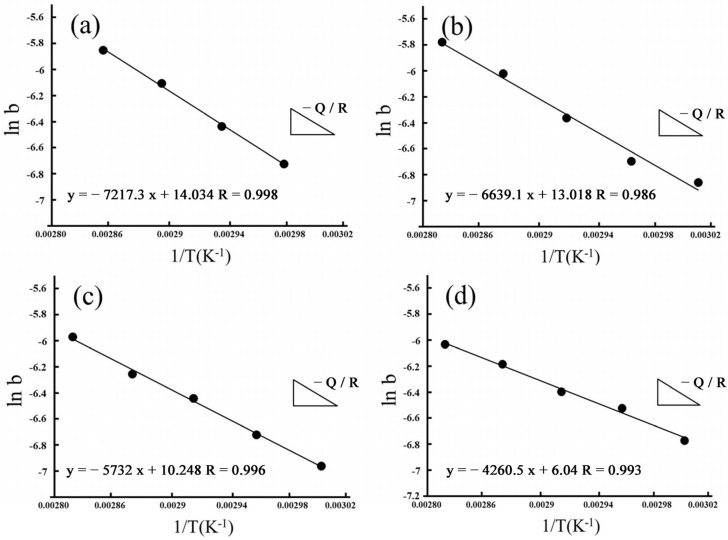
Apparent activation energies of the phase transformation for immersed degradation. Arrhenius plots linearly fitted for (**a**) 1400 °C; (**b**) 1450 °C; (**c**) 1500 °C; and (**d**) 1550 °C.

**Table 1 materials-16-07691-t001:** The Mehl–Avrami–Johnson (MAJ) equation (*n*) parameter describes the kinetics of the *t*–*m* transformation for hydrothermal degradation and immersed degradation at different sintering temperatures.

Sintering Temperature (°C)	Hydrothermal Degradation	Immersed Degradation
1400	\	3.3 ± 0.4
1450	2.3 ± 0.4	3.1 ± 0.1
1500	2.1 ± 0.1	2.2 ± 0.3
1550	2.0 ± 0.1	2.1 ± 0.4

**Table 2 materials-16-07691-t002:** Phase transformation for hydrothermal degradation and immersed degradation at different activation energies.

Sintering Temperature (°C)	Hydrothermal Degradation	Immersed Degradation
1400	\	60 kJ/mol
1450	101 kJ/mol	55 kJ/mol
1500	95 kJ/mol	48 kJ/mol
1550	86 kJ/mol	35 kJ/mol

## Data Availability

Data are contained within the article.

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
