# Peer review of "Evaluation of Feasibility on Dental Zirconia—Accelerated Aging Test by Chemical Immersion Method"

_materials, 2023, doi:10.3390/ma16247691_

Round 1
Reviewer 1 Report
Comments and Suggestions for Authors
The article is well-written, with well-conducted and relevant laboratory analyses to address the research question. However, the authors did not make clear what the innovation of the work would be. Does the study present any innovation? Have other studies already evaluated the immersion method as an alternative for the accelerated aging of zirconia? Additionally, the references used, for the most part, are from 5 years ago, which might indicate that the research topic is outdated.
Thus, the authors need to emphasize in the article's introduction what is new and important about it for it to be published
Author Response
Dear Reviewer:
It is with excitement that I resubmit to you a revised version of manuscript entitled “Evaluation of feasibility on dental zirconia accelerated aging test by chemical immersion method” for the Material. We really appreciate the reviewers’ comments and suggestions, which help us to enhance the quality of this manuscript. We have carefully revised the manuscript according to the reviewers’ suggestions. The revised parts are highlighted by yellow color in the main text. The list of changes and point-to-point as to the comments offered by reviewers are present below, and the revised text is also highlighted in the revised manuscript.
Thanks for the reviewer’s suggestion. The introduction have been carefully revised. At the same time, we have cited the recent literature.

Reviewer 2 Report
Comments and Suggestions for Authors
This is an interesting study, however, there are some comments as follows
1. Please clarify the clinical relevance and clinical simulation in case of degradation by chemical immersion
2. In case of increasing of monoclinic phase after immersion deflation, what is the consequence effect to zirconia, it would be nice if this explanation should be appear in discussion.
3. In the Page 2 Line 95, author should apply passive voice instead of using " We used commercially...... " and also others in text
Comments on the Quality of English LanguageIn the Page 2 Line 95, author should apply passive voice instead of using " We used commercially...... "
Author Response
Dear Reviewer:
It is with excitement that I resubmit to you a revised version of manuscript entitled “Evaluation of feasibility on dental zirconia accelerated aging test by chemical immersion method” for the Material. We really appreciate the reviewers’ comments and suggestions, which help us to enhance the quality of this manuscript. We have carefully revised the manuscript according to the reviewers’ suggestions. The revised parts are highlighted by yellow color in the main text. The list of changes and point-to-point as to the comments offered by reviewers are present below, and the revised text is also highlighted in the revised manuscript.
Thanks for the reviewer’s suggestion. We re-optimized this part in the revised manuscript.

Reviewer 3 Report
Comments and Suggestions for Authors
This paper introduces an accelerated test method to assess low-temperature degradation (LTD) kinetics of Y-TZP. The proposed accelerated test method, immersed degradation, is presented as a more convenient and less costly method to the alternative, hydrothermal degradation, since it does not require sophisticated equipment such autoclaves. It is always worthy to explore easier methods to assess similar outcomes, as long as the results can be trusted.
With that, I have the following two comments:
1- Results showed different activation energies for the two methods for the two reasons you explained, temperature and acidity. It would be impactful to develop a conversion system that equates the two approaches. In other words, to get the same degradation predicted by the hydrothermal degradation method, one would run the immersed degradation method at what known temperatures, times and acidity.
2- The 673 hours and 500 hours cited in the bottom paragraph of page 6 are incorrect. I believe you meant 28 hours instead of 672....etc.
Author Response
Dear Reviewer:
It is with excitement that I resubmit to you a revised version of manuscript entitled “Evaluation of feasibility on dental zirconia accelerated aging test by chemical immersion method” for the Material. We really appreciate the reviewers’ comments and suggestions, which help us to enhance the quality of this manuscript. We have carefully revised the manuscript according to the reviewers’ suggestions. The revised parts are highlighted by yellow color in the main text. The list of changes and point-to-point as to the comments offered by reviewers are present below, and the revised text is also highlighted in the revised manuscript.
Thanks for the reviewer’s suggestion. the immersed degradation method went through 28 days, that means 672 h.

Reviewer 4 Report
Comments and Suggestions for Authors
Tian et al. reported kinetics of yttria doped zirconia (Y-TZP) dental ceramic. Immersion approach was used as an alternate accelerated aging test for dental zirconia. The authors compared the amount of transferred monoclinic phase and activation energy immersion by immersion and hydrothermal methods. Information provided in the manuscript will be valuable to materials community and this reviewer recommend publishing this manuscript in the journal of Materials after addressing the minor comments below:
1. It is recommended to modify the introduction such that background and issues relating to dental zirconia should be first and the authors' approach to the problem at the last paragraph.
2. The authors should provide the XRD data after sintering and degradation as supporting information.
3. In line 29, the authors mentioned "restrain the phase transformation of zirconia because of the smaller grain size". The authors should add a brief explanation of how grain size is correlated with phase transformation.
Author Response
Dear Reviewer:
It is with excitement that I resubmit to you a revised version of manuscript entitled “Evaluation of feasibility on dental zirconia accelerated aging test by chemical immersion method” for the Material. We really appreciate the reviewers’ comments and suggestions, which help us to enhance the quality of this manuscript. We have carefully revised the manuscript according to the reviewers’ suggestions. The revised parts are highlighted by yellow color in the main text. The list of changes and point-to-point as to the comments offered by reviewers are present below, and the revised text is also highlighted in the revised manuscript.
Thanks for the reviewer’s suggestion, we have re-optimized this part in the revised manuscript.

Reviewer 5 Report
Comments and Suggestions for Authors
In this manuscript the authors investigated the influence of sinter temperature and aging regime on phase transformation and grain size of 3Y-ZrO2 ceramics for dental applications.
The study is of interest, though some aspects, such as the influence of sinter temperature are well known and established already, while others, such as the comparison between different low temperature degradation regimes, are more interesting.
In general the study is well conducted, but needs some revision, before it can be accepted for publishing.
please adjust the below mentioned parts:
- The introduction needs to be rewritten and reorganized, as it is confusing. The aim needs to be stated more clearly.
- what was the initial grain size of the starting powder of the 3Y-TZP?
-Please describe in more detail: how was the grain size determined? how was the sample preparation for SEM imaging (only polishing, etching, thermal etching, coating with gold?, etc.)?
- XRD measurement and analysis needs to be described in more detail: what powder diffraction files did you use for the analysis? how did you fit the spectra?
Comments on the Quality of English LanguagePlease check the introduction with a native speaker, it should be reorganized.
Author Response

(The authors gave the same response as above.)

Round 2
Reviewer 2 Report
Comments and Suggestions for Authors
No need for further correction
Author Response
Dear Reviewer:
It is with excitement that I resubmit to you a revised version of manuscript entitled “Evaluation of feasibility on dental zirconia accelerated aging test by chemical immersion method” for the Material. We really appreciate the reviewers’ comments and suggestions, which help us to enhance the quality of this manuscript. We have carefully revised the manuscript according to the reviewers’ suggestions. The revised parts are highlighted by yellow color in the main text. The list of changes and point-to-point as to the comments offered by reviewers are present below, and the revised text is also red highlighted in the revised manuscript.

Reviewer 5 Report
Comments and Suggestions for Authors
Thank you for the revision of the manuscript, it is greatly improved.
I have one concern left regarding this study: the authors describe the initial particle size of the Zirconia powder as 15-25 µm, but the resulting grain size after sintering is ranging between 215 and 555 nm, so 1/100 of the size of the starting powder. I assume the starting powder was a granule, with a primary grain size below 100 nm? This should be noted as well in the section 2.1. specimen preparation.
Otherwise it doesnt make sense to have much smaller grains after sintering.
The rest of the manuscript is well done, and logical and the topic is of interest.
Comments on the Quality of English LanguageLanguage is improved.
Author Response
Dear Reviewer:
It is with excitement that I resubmit to you a revised version of manuscript entitled “Evaluation of feasibility on dental zirconia accelerated aging test by chemical immersion method” for the Material. We really appreciate the reviewers’ comments and suggestions, which help us to enhance the quality of this manuscript. We have carefully revised the manuscript according to the reviewers’ suggestions. The revised parts are highlighted by yellow color in the main text. The list of changes and point-to-point as to the comments offered by reviewers are present below, and the revised text is also red highlighted in the revised manuscript.
Thanks for the reviewer's suggestion. As the reviewer mentioned, the definition of particle size is different from that of grain size. The particle size describes the size of zirconia powder particles, in which may exist several grains. After sintering, a highly uniform microstructure consisting of grains will occur. The modifications in the revised manuscript (red highlighted) are added to avoid the reader’s confusion.
